# Chitosan-Based Nanomaterial as Immune Adjuvant and Delivery Carrier for Vaccines

**DOI:** 10.3390/vaccines10111906

**Published:** 2022-11-11

**Authors:** Xiaochen Gong, Yuan Gao, Jianhong Shu, Chunjing Zhang, Kai Zhao

**Affiliations:** 1Institute of Nanobiomaterials and Immunology, School of Pharmaceutical Sciences & School of Life Science, Taizhou University, Taizhou 318000, China; 2School of Medical Technology, Qiqihar Medical University, Qiqihar 161006, China; 3College of Life Sciences and Medicine, Zhejiang Sci-Tech University, Hangzhou 310018, China; 4Zhejiang Hom-Sun Biotechnology Co., Ltd., Shaoxing 312366, China

**Keywords:** chitosan, chitosan-based nanoparticles, vaccine adjuvant, delivery system

## Abstract

With the support of modern biotechnology, vaccine technology continues to iterate. The safety and efficacy of vaccines are some of the most important areas of development in the field. As a natural substance, chitosan is widely used in numerous fields—such as immune stimulation, drug delivery, wound healing, and antibacterial procedures—due to its good biocompatibility, low toxicity, biodegradability, and adhesion. Chitosan-based nanoparticles (NPs) have attracted extensive attention with respect to vaccine adjuvants and delivery systems due to their excellent properties, which can effectively enhance immune responses. Here, we list the classifications and mechanisms of action of vaccine adjuvants. At the same time, the preparation methods of chitosan, its NPs, and their mechanism of action in the delivery system are introduced. The extensive applications of chitosan and its NPs in protein vaccines and nucleic acid vaccines are also introduced. This paper reviewed the latest research progress of chitosan-based NPs in vaccine adjuvant and drug delivery systems.

## 1. Introduction

Vaccines are biological products made from pathogenic microorganisms and parasites and their components or metabolites [1]. Vaccination can prevent and control the spread of infectious diseases by generating a sufficient protective immune response, which greatly reduces mortality and extends life expectancy [2]. Traditional vaccines, such as live attenuated vaccines, can elicit strong immune responses but present significant safety concerns. The safety of inactivated vaccines far superior, but their immune effects are poor [3,4,5]. Novel vaccines, such as subunit vaccines, can contain one or more protective antigens, but one of their disadvantages is low immunogenicity [6,7,8,9,10]. Nucleic acid vaccines include DNA vaccines and mRNA vaccines. DNA vaccines are relatively simple to construct and easy to produce on a large scale, but they have low immunogenicity due to the low rate of transfection and subsequent protein expression [11,12,13]. mRNA vaccines can be directly translated into the cytoplasm without concern for insertional mutagenesis and can be transiently expressed, possessing high transfection efficiency, fast production, relatively low cost, and versatility regarding vaccine design [11,14]. However, they are unstable, prone to degradation in the body, and lack an effective delivery vehicle [15,16,17]. Based on the characteristics of various vaccines, we summarize the advantages and disadvantages of several existing vaccines in Table 1.

Traditional vaccination mainly relies on two specific types of microbiological preparations (attenuated and inactivated) to develop vaccines. Advances in vaccine technology fields have provided alternative methods of developing safer and improved vaccines, such as subunit vaccines, recombinant vaccines, and nucleic acid vaccines. Subunit vaccines and nucleic acid vaccines, as emerging platforms, have low toxicity and a long-expression time but also suffer from low immunogenicity and low efficacy. Therefore, there is an urgent need to develop new vaccine adjuvants and delivery systems to increase the enormous potency of vaccines. Nowadays, the research and development of nano-biomaterials and the advancement of molecular biotechnology play important roles in the rational design of vaccines and in our understanding of the molecular mechanism of immune response [2,7].

Chitin is obtained from the exoskeletons of crustaceans and insects and is also a significant component of the bodies of fungi, yeasts, algae, and other lower plants’ cell walls [20,21]. Chitosan is the product of the removal of part of the acetyl group from the natural polysaccharide chitin ((1-4)-2-acetamido-2-deoxy-β-d-glucan). Chemically, chitosan (α-(1-4)-2-amino-2-deoxy-β-d-glucan) is the second most abundant polysaccharide in nature, which is composed of glucosamine and N-acetyl-d-glucosamine linked with a β-1-4-glycosidic linkage [22]. Chitosan has numerous properties, such as low-toxicity, biodegradability, and antibacterial activity, and it has been extensively studied for its biomaterial properties and use in various areas such as drug delivery, cosmetics, vaccines, antibacterial agents, and antifungal procedures [20,23]. However, because of chitosan D-glucosamine residue and a high degree of crystallinity in intermolecular and intramolecular hydrogen bonds, chitosan can only be dissolved in most diluted acids (hydrochloric acid, nitric acid, and other inorganic acids or most organic acids), and it is insoluble in neutral and alkali solutions, which limits its applications [20,22].

Therefore, it is important to improve the water solubility of chitosan. To overcome this shortcoming, researchers have proposed various methods to modify chitosan to improve its solubility. At present, the common method is to chemically modify chitosan using, for example, quaternization, alkylation, phosphorylation, acylation, etc. Modified chitosan maintains its desirable properties; moreover, it also has good water solubility due to the loss of its crystallization ability resulting from the hydrogen bond in the amino position. The increase in the positive charge of the modified chitosan provides it with good mucoadhesive and antibacterial properties. In addition, its cytotoxicity is reduced, and its immunogenicity is enhanced [24,25,26].

Currently, chitosan and its derivatives have been widely researched due to their significant characteristics [27,28,29]. Numerous studies have demonstrated that they are promising vaccine adjuvants and delivery vehicles due to their high positive charge and their ease in crossing biological barriers, which can effectively bind drugs and cells [30,31,32,33,34,35,36]. For example, compared with an avian influenza virus vaccine loaded with chitosan, an avian influenza virus vaccine encapsulated in chitosan NPs can stimulate lymphocyte proliferation [37]. A chitosan-derived-nanoparticle-loaded Newcastle disease virus (NDV) vaccine can also better enhance mucosal immune responses and antibody immune responses compared with the traditional NDV vaccine [38]. *N*-2-Hydroxypropyl trimethyl ammonium chloride chitosan (N-2-HACC) and *N*, *O*-carboxymethyl chitosan (CMCS) are two of the most effective chitosan derivatives. To further improve chitosan derivatives properties, our group synthesized N-2-HACC/CMCS NPs that retain chitosan’s desirable properties and are soluble in common organic solvents and water. The chitosan derivative’s NPs were used as adjuvants and delivery carriers for vaccines that protect against Newcastle disease and infectious bronchitis. The results showed that N-2-HACC/CMCS NPs can induce higher levels of anti-NDV IgG and anti-IBV IgG antibodies and higher levels of IL-2, IL-4, and IFN-γ [39,40,41]. Poly-ε-caprolactone (PCL)/chitosan NPs, an adjuvant for the hepatitis B surface antigen (HBsAg), can promote a cellular immune response against HBsAg, characterized by the production of IFN-γ and IL-17 [42]. Chitosan nanoparticles containing fusion protein (Hspx-PPE44-EsxV; HPE) and resiquimod adjuvant (HPERC) employed as a vaccine in BALB/c mice could produce significant amounts of IFN-γ, IL-17, and IgG2a [43].

Here, we focus on the potential of chitosan and its derivatives as adjuvants and delivery vehicles, and, subsequently, we will elucidate their preparation methods, immune mechanisms, and applications in detail for vaccine development.

## 2. Vaccine Adjuvant and Delivery System

An adjuvant is an essential component of a vaccine, and the delivery system is designed to improve vaccines’ effectiveness by delivering antigens to target organs or cells [44]. Ideal vaccine delivery systems can not only deliver and protect antigens but also act as adjuvants [45,46,47].

### 2.1. Vaccine Adjuvant

The vaccine adjuvants in development or use mainly include aluminum salts, oil emulsions, saponins, immune-stimulating complexes, liposomes, microparticles, nonionic block copolymers, polysaccharides, cytokines, and bacterial derivatives. For example, liposomes can target immune cells, cause lysosome escape, and promote antigen cross-presentation, thereby greatly improving vaccine efficacy; polysaccharides can activate the complement system and enhance the function of macrophages and natural killer cells to enhance the host’s immune defense; aluminum adjuvant agents have depot effects and prophagocytic effects and activate the pro-inflammatory NLRP3 pathway, stimulating innate and adaptive immune responses and activating the complement system; and saponins enhance CD4+ T cell-mediated immune and antibody responses and can also cause antigen cross-presentation. Details of the classification and mechanisms of vaccine adjuvants are shown in Table 2.

### 2.2. Vaccine Delivery System

Vaccine delivery systems, including viral vectors, polymeric microparticles/NPs, liposomes, and other microparticles or NPs, can protect the antigens encapsulated in NPs from degradation. A vaccine delivery carrier mainly includes viral vectors and non-viral vectors [72,73,74,75,76].

#### 2.2.1. Viral Carrier for the Delivery of Vaccine

COVID-19 is a major infectious disease that emerged in 2019. Due to the rapid mutation rate of the new coronavirus, vaccination for the infection’s prevention and treatment is integral. The high level of antigen expression produced by viral vectors lays the foundation for modern vaccine development. Viral vectors have high transduction efficiency and a wide host range [77,78,79]. However, their use is limited by toxicity and safety concerns [80,81]. Moreover, large-scale optimization is not possible due to concerns regarding reproducibility, immunogenicity, potential carcinogenicity, and the inflammation of viral vectors [82,83,84]. The viral vectors for improving the immunogenic effect of COVID-19 vaccines include adenovirus vectors, vaccinia virus vectors, measles virus vectors, rhabdovirus vectors, influenza virus vectors, and lentiviral vectors. Among them, adenoviral vectors are the most commonly used and effective, and other viral vectors are in clinical development [77,79].

Vaccines based on retroviral vectors are stably integrated into the host genome with high gene transduction efficiency. However, once infected, the probability of eradicating the virus is very low. There are studies suggesting that simple, retrovirus-based vectors could be used in gene therapy in the future [85,86]. Vaccines based on herpes simplex virus vectors can enhance the cellular, and humoral immune responses. Herpes simplex virus vaccines are rarely used in clinical practice, though one study pointed out that the combination vaccine method can improve its treatment effectiveness. However, it is highly cytotoxic [75,87,88]. The adenoviral vector vaccine induces moderate innate immunity and is highly thermostable. However, one of its disadvantages is that it can cause acute liver toxicity. It is used to prevent infectious diseases, including the Ebola virus disease, acquired immunodeficiency syndrome (AIDS), and COVID-19 [89,90,91]. Vaccines based on lentiviral vectors induce durable humoral immunity but are prone to insertional mutagenesis and have low viral titers, which easily degrade. The adenoviral vector vaccine can be used effectively in preclinical and clinical settings as a vector for gene therapy or vaccine applications (Table 3) [92,93].

#### 2.2.2. Nonviral Carriers for the Delivery of Vaccines

The nonviral carriers used for the delivery of vaccines are easy to prepare and have low toxicity, which has attracted widespread attention. Figure 1 shows the structures of nonviral vectors that have been frequently studied in recent years [76,95,96]. Chitosan-based delivery systems have been shown to have mucoadhesive properties, and they have the unique ability to open tight junctions between natural portal epithelial cells and can promote antigen transport through another effective antigen delivery pathway, thereby stimulating powerful immune responses [22].

Chitosan is a promising nonviral delivery vector because of its biocompatibility, biodegradability, low immunogenicity, and ease of manufacture. It can effectively enhance both humoral and cellular immunity, and even mucosal immune responses. Chitosan nano-vaccines are efficient carrier systems for pIL-12, and combined with HBsAg, they can induce higher levels of anti-HBsIgG, IgG1, and IgG2b; promote the maturation and presentation of dendritic cells; and enhance HBV-specific CD8+ and CD4+ T lymphocyte responses [97]. The *N*-(2-hydroxy-3-trimethyl ammonium) propyl chitosan NPs used as adjuvants for anthrax vaccine adsorption strongly stimulate humoral immunity. By using it, the level of anti-protective antigen titers in mice were significantly increased, and the survival rate was significantly improved [98]. It was difficult for macrophages to absorb *Salmonella typhi* porin; however, it was significantly absorbed by macrophages and increased the release rate of TNF-α and IL-6 after being combined with chitosan [99]. When chitosan was used in combination with IFN-β, IgG, and IgA antibodies, mucosal IgA antibodies and antitoxin titers were all stronger than those using IFN-β alone as a mucosal adjuvant [100].

Chitosan-based NPs have advantages as nonviral carriers for the delivery of vaccines [101]. Compared with monolayer-shaped liposome NPs, chitosan-based NPs have larger shapes due to the aggregation of small particles, which are more favorable for antigen delivery. For example, chitosan and liposomal NPs used as auxiliary co-delivery agents can enhance the production of IgG-class antibodies compared to liposomal NPs [102]. Chitosan-based NPs can deliver antigens and provide noninvasive routes of administration, such as oral, nasal, muscular, and ocular pathways.

The pathogens delivered by chitosan and its nanocomplexes via ocular, oral, intranasal, and intramuscular routes in recent years are shown in Table 4. It has been shown that the opening of tight junctions between chitosan cells favors the paracellular transport of hydrophilic macromolecular drug macromolecules such as peptides and proteins [103,104]. Oral vaccines are the easiest route of administration with the least risk of contamination. However, due to the influence of the gastrointestinal environment, protein antigens are inactive and have poor absorption properties as well as low bioavailability. Chitosan mesoporous silica NPs have potential as oral vaccine carriers. The NPs effectively protect proteins from degradation due to pepsin and trypsin. Plasmid DNA containing lymphovirus genes was encapsulated in chitosan-based microspheres, which also prevented the hydrolytic denaturation of DNA vaccines in the gastrointestinal tract [104,105]. The cross-presentation of foreign antigens is a prerequisite for cytotoxic T lymphocyte responses. Chitosan/calcium phosphate nanosheets are used as vaccine carriers to effectively cross-present exogenous antigens. The water solubility of chitosan greatly limits its application as a delivery vehicle [106]. An antigen-coated with n-phosphochitosan not only solves the problem of water solubility but also significantly improves the level of the antigen-specific immune response, and it can be used as an antigen carrier for immune prevention and treatment [107].

## 3. Preparation of Chitosan-Based NPs for Vaccine Delivery

Chitosan-based NPs can be prepared for vaccine delivery using both physical and chemical methods. Although both methods have pros and cons, the physical methods used to yield particles are preferred over chemical crosslinking methods because the proteins are chemically modified with crosslinking agents, proteins are degraded in organic solvents, and it is difficult to remove the unreacted crosslinkers from the formulations. The ion-crosslinking method uses the positive charge of chitosan and other negatively charged substances as part of an electrostatic effect in order to prepare NPs. The cross-linking agent used has low toxicity and avoids the toxicity caused by the use of organic solvents [142]. Polyelectrolyte recombination is caused by the interaction between polyelectrolytes with opposite charges. This method also does not use a chemical crosslinking agent, and its process is simple, but its stability is easily affected by pH, thus affecting the charge and size of the NPs [143,144]. The desolvation method uses the insolubility of chitosan in an alkaline medium to form sediment. Its nanoparticle formation rate is high, but the nanoparticles produced are large [145]. The emulsification method is based on the method of removing the organic solvent and comprises techniques such as the emulsification solvent evaporation method and the emulsification solvent diffusion method [146]. This method is simple, but if the organic solvent is not completely removed, it will affect cells and tissues. In addition, the selection of the surfactant has a great impact on the stability of the system [147,148,149,150]. The spray-drying method is used to dissolve and disperse chitosan in an acidic solution, and then spray it in a hot gas flow to rapidly evaporate the solvent into gas [151,152]. This method is simple, but the size of the NPs is not easy to control. The covalent cross-linking method [153,154] uses the formation of covalent bonds between chitosan NPs and cross-linking agents. The cross-linking agents used in the preparation process are highly toxic. The reverse micelle method involves grafting chitosan and lipophilic substances, with the graft copolymer dispersed in an organic solvent, which is a process that is simple to carry out. However, the use of some organic solvents has certain toxic effects on cells [155]. Here, various antigen formulation techniques are discussed, such as ionic crosslinking, polyelectrolyte complexation, spray drying, reverse micellization, covalent crosslinking, and so on. Ion crosslinking is one of the most widely used methods for the preparation of chitosan NPs (Table 5). Each method, of which ionic crosslinking method is the most commonly used, has advantages and disadvantages. Schematic diagrams of these methods and preparation procedures can be found in the literature [156].

Moreover, many factors can influence the effects of vaccine adjuvants and delivery systems. For example, the size of the NPs will affect their absorption in the body, the particle size will affect the transmembrane transport ability of the NPs, and the positive charge characteristic of chitosan will easily interact with negatively charged surfaces (such as mucous membranes), thereby influencing the immune effect [157]. Among the current methods for preparing NPs, some are toxic to cells due to their use of organic solvents. The removal of organic solvents is very important. In addition, the selection of surfactants also has a great influence on the particle size, morphology, and stability of NPs, so the research and development requirements of surfactants are becoming more stringent in order to enable the application of nanomaterials in different fields. In the development of chitosan derivatives, using different chemical reagents to synthesize NPs will produce different degrees of toxicity. In addition, preclinical studies need to evaluate the safety of chitosan-based nanocarriers. Therefore, an appropriate synthesis method must be selected according to the different applications of the NPs, and more non-toxic NPs should be prepared [158].

## 4. Immune Mechanism of Chitosan and Its Nanocomposites as the Vaccine Adjuvants/Delivery System

The immune enhancement mechanism of chitosan NPs mainly includes its effect on the antigen, enhancing the targeting of antigen-presenting cells, and provoking macrophages to secrete related inflammatory factors and regulate Th1/Th2 propensity to regulate the immune response. Since antigens are negatively charged, chitosan NPs readily bind to antigens for the purposes of storing and slowing antigen release and activating antigen-presenting cells to make Th1/Th2 responses more balanced for use as vaccine adjuvants/delivery systems [159,160,161]. NPs can enhance cell-mediated immune responses as well as humoral immune responses. In addition, they induce superior uptake via antigen-presenting cells because of their positive charge and their ionic interactions with negatively charged cell membranes (Figure 2) [161,162,163]. The chemical modification of chitosan can also enhance the immune effects of vaccines. For example, N,N,N-trimethyl chitosan can induce the maturation of DCs and improve the cell uptake of vaccine antigens, while HTCC (or HACC) can induce the Th2 type response to reduce allergic reactions. N-phosphonium chitosan stimulates Th1 and Th2 responses; however, the secretion of IFN-γ and IL-12 is downregulated compared to the Freund’s adjuvant. Glycol chitosan induced stronger immune responses than chitosan NPs after nasal immunization, which suggests the GC NPs have the potential to become promising immune-related materials for mucosal administration [30].

The successful activation of antigen-presenting cells is a key factor in effective adjuvants/delivery systems. Chitosan allows antigen-presenting cells to deliver immunomodulatory molecules without causing adverse immune responses. Chitosan can promote dendritic cell maturation and enhances antigen-specific helper T cell 1 (ThA) responses by participating in the sting–cGAS pathway [67]. Ovalbumin (OVA)/Kirdran sulfate/O-HTCC NPs induce the activation and maturation of antigen-presenting cells, stimulate the proliferation and differentiation of lymphocytes, and induce higher levels of OVA-specific antibodies in mice [164]. Studies have also shown that O-Sulfate/o-quaternary chitosan NPs can efficiently enter swine macrophages, escape from lysosomes, promote cytokine production in immune cells, and activate PI3K/AKT and MAPK pathways, as well as promote lymphocyte proliferation, dendritic cell phenotypes, and functional maturation. Furthermore, chitosan can enhance the immunogenicity of the porcine circovirus vaccine against infectious pathogens that are associated with the over-expression of TLR2 and TLR4 in macrophages [165].

Chitosan and its nanocomposites can induce Th1/Th2 immune responses [166]. The Th1 immune response is a pro-inflammatory response produced by Th1 cells against intracellular microorganisms such as bacteria and viruses. The Th2 immune response is an anti-inflammatory response generated by Th2 cells against extracellular microorganisms. An excessive Th1 immune response can lead to an autoimmune response that then leads to uncontrolled tissue damage. The Th2 immune response can counteract the Th1 immune response. A balanced Th1/Th2 response is particularly important for the organism. Chitosan can promote balance in Th1/Th2 responses [25]. For example, the murine intranasal immunization of *Brucella* protein-loaded chitosan NPs produces Th1- and Th2-related antibodies. In one study, murine ovalbumin was encapsulated in chitosan NPs, which significantly promoted murine Th1 (IL-2 and IFN-γ) and Th2 (IL-10) cytokine production and triggered a balanced Th1/Th2 response [167,168]. Chitosan and c-di-GMP combinations skew cytokine responses toward a more balanced Th spectrum; the complexation of chitosan and heparin to prepare nano-vaccines induces strong immunity to Th1 responses, in addition to co-strong Th2 response activation; and the presence of chitosan redirects immune responses to a more balanced Th1/Th2 pathway compared to alum adjuvant [168]. Ghendon et al. used chitosan as an adjuvant of an H5 inactivated influenza vaccine. The results showed that an intramuscular injection could significantly improve antibody titer and protection efficiency and promote the increase in the number of CD3, CD3/NK, I-AK (MHCII), and H-2Db (MHCI) cells [169].

## 5. Application of Chitosan and Its Nanocomposites in Vaccine Delivery

### 5.1. Application in Protein Vaccine Delivery

Protein-encapsulated chitosan and its nanocomposite particles are being explored for a variety of biomedical applications. The size, morphology, and surface charges of NPs can modulate interactions with immune cells, which have a great influence on the immune response [170,171]. Chitosan NPs stimulate immune cells and have chemically modifiable active functional groups, all of which impart multiple advantages in vaccine delivery [95]. Nanoparticle carriers are used to transport antigenic proteins in vivo and can avoid proteolysis or the degradation of antigens. The alginate-chitosan-coated LDH nanocomposite (ALG–CHT-LDH) can effectively protect antigens from acidic degradation, significantly enhancing protein attachment and the internalization of proteins in enterocytes and macrophages, and it can be used as an oral protein vaccine carrier [172]. Oral immunization with recombinant bovine serum albumin conjugated to mannosylated chitosan NPs elicits strong systemic IgG antibody and mucosal IgA responses. The prepared NPs also reduce the effects of the gastric environment [173]. The positive charge on the nanoparticles’ surface also enhances immunogenicity. Kaneko et al. used chitosan hydrochloride to induce pneumococcal protein’s antigen expression. Negatively charged chitosan NPs generally exhibit lower immunogenicity compared with positively charged chitosan NPs, which may be related to differences in the surface charge [174].

Chitosan and its nanocomposites play an important role in the development of vaccines for animal diseases. Chitosan NPs efficiently deliver *Salmonella* subunit antigens via the oral route and increase the expression levels of TLRs, Th1, and Th2 cytokine mRNA in chicken immune cells [175]. *Brucella* is one of the most common zoonotic diseases. Compared with live attenuated vaccines, recombinant protein vaccines and subunit vaccines have been the focus of research in recent years regarding the prevention of brucellosis. Mannosylated chitosan NPs (MCN)—loaded with FliC protein as a targeted vaccine delivery system to develop subunit vaccines against brucellosis—could significantly enhance the immunogenicity of FliC protein’s highly specific IgG response and produce high levels of IFN-γ and IL-2 [176].

### 5.2. Application in Polypeptide Vaccine Delivery

Peptide-based vaccines have attracted enormous attention in recent years as one of the most effective disease prevention methods. However, peptide epitopes are weakly immunogenic and unstable. Nanoparticle adjuvants or delivery systems can induce highly durable immune responses against weakly immunogenic peptides [177]. A polyglutamic acid–peptide conjugate and trimethyl chitosan were used to prepare a nanoparticle peptide vaccine that could effectively improve immunogenicity [178]. The conjugation of the unstable P10 peptide to chitosan NPs promoted immune responses with Th1 and Th2 [179]. A highly efficient dendritic-cell-targeted vaccine delivery system was developed by coupling a targeting peptide (TP) with chitosan. Compared with chitosan NPs, the TP-coupled chitosan NPs (TPC-NPs) have an increased ability to target DCs and have an enhanced immune function [180]. Table 6 briefly summarizes the application of chitosan and its nanocomposite particles in the delivery of protein vaccines and polypeptide vaccines in recent years.

### 5.3. Application in Nucleic Acid Vaccine Delivery

#### 5.3.1. DNA Vaccines

DNA vaccines are a promising method for the treatment and prevention of diseases because of their safety (carrying only specific antigenic genetic information), stability, and the ease of their large-scale preparation and production. The coformulation of a gene delivery vehicle with an immunostimulant enhances the therapeutic efficacy of DNA vaccines [191]. Chitosan NPs as a nano-delivery system induce high levels of IFN-γ and IL-4 and enhance the effect of the DNA vaccine, and the effect is better than that of chitosan and IL-12 alone [192].

The COVID-19 pandemic has affected nearly 210 million people around the world. A variety of COVID-19 vaccines have been approved for human applications, including live attenuated and inactivated virus and protein subunit vaccines. The immune response to the intranasal delivery of the SC2-spike DNA vaccine, transported on a modified gold–chitosan nanocarrier, demonstrates strong and consistent antibodies (IgG, IgA, and IgM) and the effective neutralization of pseudoviruses expressing the S proteins of different SC2 variants (Wuhan, beta, and D614G) [193]. The pVAX1-constructed DNA vaccine can improve the survival rate of carp, goldfish, and other fish against *Aeromonas hydrophila* infection [194]. In addition, PLGA–chitosan NPs induce adaptive mucosal immunity in fish, and the mRNA expression levels of IgM, IgD, and IgT are significantly upregulated compared to pDNA–PLGA–NPs [195].

Infectious bronchitis (IB) is an important infectious disease in poultry. Chitosan saponin NPs can encapsulate monovalent and bivalent DNA vaccines developed based on the S1 glycoprotein of classical and variant IBV strains (M41 and CR88), resulting in high antibody titers as well as enhanced immunity ingenuity and protection in chickens [148]. Importantly, in comparison with inactivated vaccines, multiple booster vaccinations are not required to control poultry-infectious bronchitis. Our group prepared *O*-2′-hydroxypropyl trimethylammonium chloride chitosan containing NDV F gene plasmid DNA and the C3d6 molecular adjuvant, which induced higher levels of anti-NDV IgG and sIgA antibody titers and significantly promoted lymphocyte proliferation, producing higher levels of IL-2, IL-4, IFN-γ, CD4^+^, and CD8^+^ T lymphocytes [41]. The intranasal delivery of *O*-2′-HACC/pFDNA can enhance humoral, cellular, and mucosal immune responses and protect chickens against highly toxic NDV compared to commercial live attenuated NDV vaccines. The applications of several chitosan-based complexes in DNA vaccines are listed in Table 7, which indicates the great potential of chitosan and its NPs in DNA vaccines.

#### 5.3.2. RNA Vaccines

The development of delivery systems will address many of the problems inherent to in vivo RNA delivery to improve the key efficacy of RNA drugs. mRNA vaccines are more potent and specific than protein/peptide vaccines in stimulating humoral and cellular immune responses [203]. Studies have shown that mRNA vaccines can cross the mucosal barrier and rapidly adapt to future outbreaks of new infectious diseases [204]. However, their degradation rate in vivo is fast, and their transfection efficiency is low. mRNA is also easily degraded by nucleases during delivery. The most commonly used approach is to encapsulate mRNA in cationic NPs; for example, chitosan and its NPs can deliver immunogenic antigens to antigen-presenting cells [205].

Chitosan–alginate-gel-scaffold-mediated mRNA liposomes enable T cell proliferation and high levels of IFN-γ compared to the injection of naked mRNA or mRNA liposomes. The nature of RepRNA uptake by DCs is inefficient. The encapsulation of RepRNA into chitosan NPs has demonstrated its efficient delivery to DCs and the induction of immune responses in vivo [206]. Steinle et al. incorporated synthetic humanized Gaussian luciferase (hGLuc) mRNA into a chitosan–alginate hybrid hydrogel. mRNA-containing hydrogels can be used to deliver locally and continuously synthesized mRNA to cells within weeks with a potency of up to 21 days [207]. Mannosylated-chitosan-modified ethosomes loaded with mRNA can efficiently transfect DCs and induce their maturation. The results of one study showed that mannosylated-chitosan-modified liposome has a good targeting ability and can protect mRNA from being degraded. Its transdermal ability is a good vehicle for the targeted delivery of mRNA. Compared with traditional oral or injectable mRNA vaccines, transdermal immunization is also becoming increasingly extensive. Chitosan-modified vaccines can allow antigen molecules to easily enter the epidermis to activate dendritic cells [208].

mRNA vaccines are highly effective and safe for preventing COVID-19 according to clinical trials. mRNA-1273 and BNT162b2, two COVID-19 mRNA vaccines, are approved for use in the market. Moreover, four other COVID-19 mRNA vaccines are also in clinical trials, including ARCoV (Abogen, China), CvnCoV (CurVac, Germany), ARCT-021 (Arcturus, USA), LNP-nCoVsaRNA (Imperial College London, England), and ChulaCoV19 mRNA vaccines (Chulalongkorn University, Thailand) [209,210]. As we all know, an efficient in vivo delivery is critical for mRNA vaccines to achieve prophylactic relevance. Lipid nanoparticles (LNPs) are the most widespread platform and have been shown to present the best clinical outcomes in mRNA delivery. LNPs mainly consist of ionizable lipids, cholesterol, phospholipids, and polyethylene glycol (PEG) lipids. The ionizable lipids contain amine groups, which become cationic at a low pH and, therefore, can efficiently complex negatively charged mRNA. Once delivered into the endosomes of host cells, they are thought to be ionized negatively again upon acidification, which helps to induce hexagonal phase structures and, finally, facilitates the endosomal escape of mRNA into the cytoplasm. Phospholipids play a structural role in LNPs, assisting the formulation of LNPs and the disruption of the lipid bilayer to promote the endosomal escape of mRNA. Cholesterol serves as a stabilizing element in LNPs. Lipid-anchored PEGs dominantly deposit on the LNP surface as barriers to sterically stabilize the LNP and reduce nonspecific binding to proteins [211]. This biopolymer can be a potential biomacromolecule to be employed against the SARS-CoV-2 *via* the preparation of antiviral vaccines and adjuvants. We hope more research will focus on this excellent nanomaterial. Table 8 lists the applications of several chitosan based complexes in RNA vaccines. 

## 6. Future Perspectives

Although the safety of subunit vaccines and nucleic acid vaccines has been greatly improved, their immunogenicity is low. The development of adjuvants and delivery systems is critical for the improvement of these vaccines. Natural polymer chitosan and its NPs have immunoreactive and mucosal adhesion properties, and they can also improve the stability and immunogenicity of the antigen. Since they stimulate the immune response and can achieve antigen-specific release through targeted delivery, chitosan and its nanocomposites have great potential as delivery systems [41,212,213]. At the same time, the application of chitosan and its complexes in nucleic acid vaccines is also increasing. Using chitosan and its nanoparticle complexes to interact with negative charges in capsid proteins via electrostatic interactions in order to cause structural damage to the virus could potentially treat COVID-19. In addition, the direct delivery of chitosan NPs to the lungs can also help with COVID-19 drug treatments. However, the physicochemical properties of chitosan, such as deacetylation, molecular weight, and systemic pH can affect antigen delivery. As a natural polymer, the toxicological issues of chitosan NPs are still a major concern. The toxicity of all chitosan derivatives in vivo should be investigated. Pre-clinical investigations are also needed to evaluate the bioactivity and safety of chitosan-based nanocarriers. Another major concern for chitosan and its derivatives is that their immunological profiles have not been studied thoroughly. When chitosan is used with some clinical drugs, some adverse reactions will occur. For example, the positive charge of chitosan can react with lipids and bile acids; affect the absorption of vitamin A, D, E, and K; and thus enhance the anticoagulation effect of warfarin [214]. The mucosal adhesion properties of chitosan can prevent acyclovir from being transported through the gastrointestinal tract [215]. Huang et al. demonstrated that chitosan particles had a significant proinflammatory effect on lung tissue after being inhaled into the lungs [216]. Therefore, the studies of chitosan and its nanoparticle complexes are in the experimental stage and have not yet been applied in clinics.

During the synthesis of chitosan, temperature, time, and the repetition of the alkali step all affect the degree of deacetylation, which, in turn, affects the synthesis of chitosan. It is necessary to optimize the synthesis conditions of chitosan to reduce the error between batches of chitosan. Chitosan’s solubility can be increased by modifying it to form new derivatives. The delivery mechanism of chitosan and its NPs in vivo should also be further explored with the support of modern biotechnology. In conclusion, chitosan and its NPs have great potential and greater development prospects for vaccine delivery.

## Figures and Tables

**Figure 1 vaccines-10-01906-f001:**
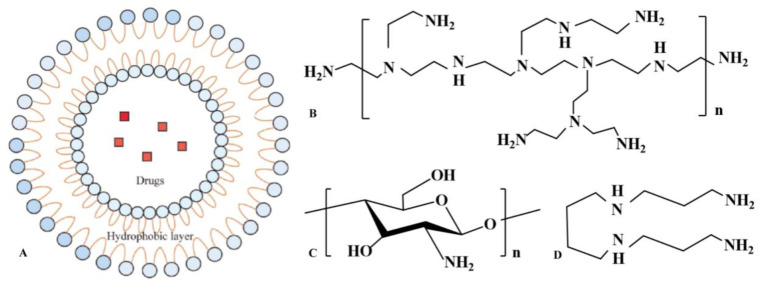
Well-studied nonviral vectors. (**A**). liposome; (**B**). polyethyleneimine; (**C**). chitosan; (**D**). spermine.

**Figure 2 vaccines-10-01906-f002:**
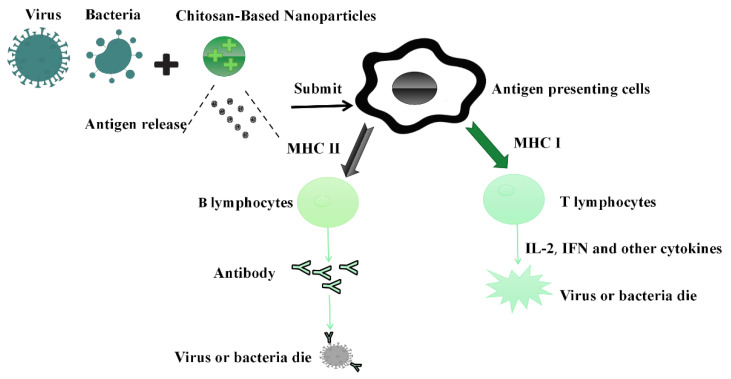
Immune mechanism of chitosan against viruses and bacteria.

**Table 1 vaccines-10-01906-t001:** Preparation, advantages, and disadvantages of existing vaccines.

Existing Vaccine	Preparation	Advantages	Disadvantages	References
Attenuated vaccines	Produced from pathogenic isolates via serial passage in embryonated, specific, pathogen-free eggs	Cause a strong immune response; long-lasting immune response; can survive the low-pH enzymatic environment of the stomach	Risk of toxic recovery in immunocompromised individuals; low safety profile	[3,4]
Inactivated vaccines	Formaldehyde inactivates live virus production	Compared with live attenuated vaccines, the safety is improved, and the production process is mature	Weak immune response; short duration; destruction or alteration of antigens; high cost; multiple vaccinations required	[4]
Subunit vaccines	Antigenic proteins from pathogens	High safety; modulated immune response; clear target antigen	Low immunogenicity	[4,5,6,7]
DNA vaccines	Transfection of DNA encoding a certain antigenic protein into animal cells	Simple to build; easy to mass-produce; high security and high stability	Cannot be used in humans; naked-pelleted DNA has low immunogenicity	[8,9,10,11]
mRNA vaccines	Transfection of RNA encoding a certain antigenic protein into animal cells	High transfection rate and simple process	Unstable; lack of effective delivery vehicles in vivo	[8,14,15,16,17,18,19]

**Table 2 vaccines-10-01906-t002:** Classification and mechanisms of vaccine adjuvants.

Classification	Subdivision	Example	Mechanism
Inorganic	Aluminum salt	Aluminum hydroxide; Alhydrogel; Aluminum phosphate; Adju-Phos; Merck aluminum; AAHSA; Nano aluminum	Prolonged interactions between antigens and immune cells; prophagocytosis; induction of Th2 immune-biased responses; activation of the pro-inflammatory NLRP3 pathway [45,48]
	Calcium salt	Calcium phosphate; Calcium phosphate NPs	Induces a more balanced Th1 and Th2 immune response [49]
	Other inorganic substances	Mn^2+^	Enhancement of immune responses through the STING pathway in dendritic cells [50]
Organic matter	Oil-in-water emulsion	MF59; AS03; AF03; SE	Induction of a Th2 immune bias response; local activation of the innate immune system [51]
	Bacterially derived molecules	Monophosphoryl lipid A (MPL); Pyranose-based lipid adjuvant (GLA); RC529; CpG; stable toxin (LT); Cholera toxin (CT); TDB; Flagellin	Activates Toll-like receptor 4; triggers cells expressing Toll-like receptor 9 (including human plasmacytoid dendritic cells and B cells); induces Th1/Th2/Th17 responses; promotes dendritic cell uptake and maturation; activates the ERK1/2 pathway via PLC-γ1/PKC signaling [52,53,54,55,56,57,58,59]
	Virus-derived molecules	Poly I: C	Activation of TLR3; melanoma differentiation-associated gene 5 (MDA-5); cellular immunity and type I interferon response [60]
	Plant-derived molecules	QS21; Delta inulin	Promotes CD4^+^ T cell-mediated immune responses; induces a more balanced Th1/Th2 response [61,62].
	Endogenous molecules	Cytokine	Activates dendritic cells [63]
	Synthesis of artificial small molecules	Imiquimod; R848; Cyclosporine	Activates TLR7/8 receptors and T cell co-stimulator CD40 [64,65]
Composite adjuvant	Aluminum-based composite adjuvant	AS04	Rapidly triggers local and transient cytokine responses to enhance humoral and cell-mediated responses, resulting in increased activation of antigen-presenting cells and promoting antigen presentation in CD4+ T lymphocytes [66]
	Emulsion-based compound adjuvant	AS02 (AS03 + MPL+QS21), GLA-SE	Activation of TLR-4 regulates immune homeostasis and promotes Th1-type immune responses [67]
	Lipid group compound adjuvant	AS01 (liposome+MPL+QS21); AS15 (AS01+CpG); ISCOMs	Effectively promotes CD4+ T lymphocyte-mediated immune response [68]
Polymer particles	CaCO_3_-LNT; Chitosan		Enhances the expression of MHC-II and CD86 in dendritic cells and increases the ratio of CD4+ to CD8+ T lymphocytes; provides a depot for the antigen to slowly release; facilitates the antigen’s targeting of immune cells; improves phagocytosis, modulation, and enhancement of the immune response induced by the antigen alone [69,70,71].

**Table 3 vaccines-10-01906-t003:** Viral vectors and their advantages and disadvantages.

Virus Vectors	Advantages	Disadvantages	References
Retroviral-vector-based vaccines	Stable integration into the host genome; high gene transduction efficiency	Once infected, the likelihood of eliminating the virus is low	[86]
Herpes simplex virus vector vaccine	Enhances cellular and humoral immunity; Induces lasting immune response.	High cytotoxicity	[75,88,89]
Adenovirus vector vaccine	Induces moderate innate immunity; high thermal stability	Cause acute hepatotoxicity	[90,91,92]
Lentiviral vector vaccine	Induces durable humoral immunity	Insertional mutagenesis; yields low viral titers; easily degradable	[93,94]

**Table 4 vaccines-10-01906-t004:** A summary of chitosan and its nanocomplexes for pathogen delivery.

Immunization Pathway	Pathogens
Intramuscular immunization	*Chlamydia* [108], *Mycoplasma* [109], Influenza virus [65], *Bacillus anthracis* [110], *Mycobacterium tuberculosis* [111], HIV [112] *Treponema pallidum* [113], Herpes virus [75], Japanese encephalitis virus [114], Cholera virus [115], Avian influenza virus [37], *Edwardsiella tarda* [116]
Intranasal immunization	*Campylobacter jejuni* [117], Foot-and-mouth disease [118], PR8 influenza virus [119], *Escherichia coli* [87], Diphtheria toxin [120], Influenza A virus [119], Newcastle disease virus [9], *Streptococcus pneumoniae* [121], Coxsackievirus [122], *Vibrio cholerae* [123]
Oral immunization	*Salmonella* [99], *Brucella* [124], Hepatitis B virus [21], Recombinant enterovirus [125], Koi herpesvirus [126], *Schisoma mansoni* [127], *Vibrio cholerae* [123], tetanus toxoid [128], diphtheria toxin [129], Schistosomiasis [130]
Ocular immunization	Cyclosporin A [131], Herpes simplex virus [132]
Subcutaneous immunization	Hepatitis B virus [133], Hepatitis B surface antigen [134], *Mycoplasma pneumoniae* [135], *E. Coli* [136], Tetanus toxoid [137], *Toxoplasma gondii histones* [138], *Echinococcus granulosus* [139], Influenza virus [37], *Leishmania* antigen [140], *Mycobacterium tuberculosis* [141]

**Table 5 vaccines-10-01906-t005:** Preparation methods of chitosan NPs.

Preparation Methods	Mechanism	Advantages	Disadvantages
Ion crosslinking [142]	Interaction between crosslinking agent and the amino or carboxyl groups of chitosan NPs.	The preparation process itself has no organic solvent; the reaction conditions are simple, mild, and controllable.	Not completely immune to gastric acid degradation; low solubility.
Polyelectrolyte complexation [143,144]	Interactions between oppositely charged polyelectrolytes.	Two- or three-step process; the equipment requirements are not demanding, and the conditions are mild.	Stability is susceptible to pH.
Desolvation [145]	Insolubility of chitosan in alkaline media.	High nanoparticle formation rate; improved physical stability.	Inhomogeneous distribution of NPs; difficulty in synthesizing smaller-sized chitosan NPs.
Emulsification [146,147,148,149,150]	The oil and water phases are emulsified; then, the solvent is removed.	Two- to three-step process; no need for sonication or homogenizers.	Causes significant toxicity to tissues or cells; poor stability.
Spray drying [151,152]	Amino groups can be protonated by acids.	Re-dispersibility; enables easier synthesis of smaller sizes compared other methods; low toxicity.	The particle size is not easy to control; the particles are irregularly shaped and sticky.
Covalent crosslinking [153,154]	Formation of covalent bonds between chitosan NPs and crosslinking agents.	Controllable drug release.	Cytotoxic.
Reverse micelle [153,155]	Trans-interaction between chitosan NPs and crosslinking agent.	One-step process; good dispersion.	Causes significant toxicity in tissues or cells.

**Table 6 vaccines-10-01906-t006:** Application of chitosan in protein vaccines and polypeptide vaccines.

Chitosan Complex	Protein/Polypeptide	Immune Mode	Target Disease
Chitosan-modified silica NPs [104]	Bovine serum albumin	Oral	Not mentioned
Chitosan NPs [175]	*Salmonella enteritidis* outer membrane proteins (OMPs) and flagellin proteins	Oral	Brucellosis
Chitosan NPs [181]	Fusogenic protein p10 of avian reovirus (ARV-p10)	Intramuscular injection	Melanoma
Mannosylated chitosan NPs [176]	FliC antigen	Subcutaneous injection	Brucellosis
*N*-2-Hydroxypropyl trimethyl ammonium chloride chitosan [39]	Porcine parvovirusVP2 protein	Intramuscular injection	Porcine *parvovirus* disease
Mannose-conjugated chitosan [182]	SwIAV antigen	Intranasal	Swine influenza
Chitosan [183]	Recombinant protein Pac	Oral	Dental caries
Chitosan polymeric NPs [179]	P10 peptide	Intranasal	Paracoccidioidomycosis
Chitosan–PLGA NPs [184]	rOmp22 peptide	Subcutaneous injection	*Acinetobacter baumannii*
Trimethyl chitosan [178]	*A. streptococcus* peptide	Intranasal	Group A *Streptococcus*
Alginate/chitosan/alginate microcapsules [185]	Probiotic expressing M cell homing peptide	Oral	Not mentioned
Alginate–chitosan–PLGA complex [186]	Ac-PLP-BPI-NH_2_-2 peptide	Subcutaneous injection	Autoimmune encephalomyelitis
Alginate and trimethyl chitosan [187]	Lipopeptide	Oral	Group A *Streptococcus*
Trimethyl Chitosan NPs [188]	Malaria antigens	Intramuscular injection	Malaria
Chitosan [189]	Receptor-binding domain (RBD) polypeptides	Subcutaneous injection	SARS-CoV-2
Chitosan-mannose NPs [190]	Recombinant Art v 1 wormwood pollen protein	Intramuscular, subcutaneous, subcutaneous injection	Allergic bronchial asthma

**Table 7 vaccines-10-01906-t007:** Application of chitosan-based complexes in DNA vaccines.

Chitosan Complex	Causative Agent	Immune Animal or Model	Immunization Mode
Chitosan–saponin [196]	Avian infectious bronchitis virus	Chicken	Intramuscular injection
Chitosan–trimeric phosphate [194]	pVAX-OMP and pVAX-hly	Carp	Oral
Polylactic-co-glycolic acid-chitosan NPs [197]	Late-onset *Edwardian* spp.	Fish	Immunization by immersion
*N*-2-Hydroxypropyl dimethylethyl ammonium chloride chitosan NPs [38]	Newcastle disease virus	Chicken	Intranasal
Mannosylated chitosan [198]	*Mycobacterium pneumonia*	Mouse	Intranasal
N-2-HACC/CMCS NPs [40]	Newcastle disease virus	Chicken	Intranasal
Chitosan NPs [199]	Herpes simplex virus	Duck	Intramuscular injection
Chitosan NPs [200]	Hepatitis B antigen	Mouse	Intranasal
Chitosan NPs [201]	*Helicobacter pylori*	Mouse	Oral
Mannosylated chitosan NPs [202]	Mouth disease virus	Guinea pig	Intranasal

**Table 8 vaccines-10-01906-t008:** Application of chitosan-based complexes in RNA vaccine.

Chitosan Complex	Causative Agent	Diseases
Chitosan NPs [205]	Influenza H9N2 HA2 and M2e mRNA	Influenza
Chitosan NPs [206]	Influenza virus	Influenza
Chitosan–alginate hybrid hydrogel [207]	Synthetic mRNAs for tissue-engineering applications	Tissue-engineering applications
Mannosylated-chitosan-modified ethosomes [208]	Tyrosinase-related protein 2	Melanoma

## Data Availability

Not applicable.

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
