# Peer review of "Chitosan-Based Nanomaterial as Immune Adjuvant and Delivery Carrier for Vaccines"

_vaccines, 2022, doi:10.3390/vaccines10111906_

Round 1
Reviewer 1 Report (New Reviewer)
The manuscript herein presented reviews the potential of chitosan for application in vaccines either as an adjuvant and a delivery carrier. The manuscript present a collection of work developed in the area in a comprehensive manner which might be of importance for researchers in the field. Nevertheless, there are a few comments and modifications that, in my opinion, are needed for this manuscript to become suitable for publication.
1. The manuscript contains a large number of tables that although compile important information. However, these table are missing some information and should be improved:
table 4 - missing references of the works described
table 5 - references are missing; the description of the methodologies and the output could be improved with a better description. The authors refer often to simple processes or small or high particle sizes but never describe what they consider as a simple process or what is the actual size or size interval of the particles.
table 6 - References are missing; please add the target disease for which polypeptide vaccine is being developed
table 7 - References are missing; the immune animal correspond to the final application of the vaccine or is the model used to test the efficacy of the vaccines?
2. In chapter 3 the authors discuss the methods to prepare chitosan NP for vaccine delivery. Although there are described several methods toxic and non-toxic, the authors conclude that 'more non-toxic NPs should be prepared'. Why is there a need for new methods? Why are the non-toxic methods described enough? Please discuss this subject in more detail.
3. In the chapter regarding the mechanism of action of chitosan it is difficult to understand if the discussion is about an adjuvant effect, with implications for the immune response or about a delivery system. Please clarify this issue. Furthermore, the authors should include information regarding the models used in chitosan vaccine studies. Where this immune data collected from in vivo models? which type of models (rodents, macaques, humans...)?
4. Regarding the effect of chitosan as an vacine adjuvant. Reading the manuscript, I wonder if this immune stimulation is due to chitosan itself or due to the chemical modifications performed. Is it possible for the author to improve a discussion regarding the effect of chemical modification on the immune stimulant properties of chitosan?
5. A critical flaw of this manuscript is the lack of information regarding the current status of chitosan use in vaccines: is there any approved vaccine using chitosan for humans/animals? Are there relevant clinical studies that should be mention?
Author Response
Please see the attachment. Thanks!

Reviewer 2 Report (New Reviewer)
The potential applications chitosan-based NPs as adjuvants for vaccines is gaining significant attention due to the unique properties of this delivery system, which make it attractive. However, lot remain unclear about this system. In this review article, the authors listed the classifications and mechanisms of action of vaccine adjuvants. In addition, the authors discussed the methods of chitosan preparation as well as its NPs. They also and explained the mechanism of action of this delivery system. Most importantly, the authors devoted significant portion of this review article to the extensive applications of chitosan and its NPs in protein vaccines, DNA, and RNA vaccines. Towards the end of the review, they discussed the latest research progress of chitosan-based NPs in vaccine adjuvant and drug delivery systems and pointed to the future potential applications of chitosan-based NP as immune adjuvant and delivery career for vaccines.
Overall, the review is well written, and it is of high relevance to the field. It highlighted most of the advantages of chitosan-based NPs as an excellent delivery system, which also can serve as suitable adjuvant for most of the protein and nucleic acid-based vaccines. At the same the authors discussed any needing improvement. The authors provided an elaborate review which encompass scientific reports since 2001. The article is well referenced and its content will contribute significantly to the field. The authors are among the leaders on this technology. Lastly, in my opinion, it will attract readers in the field. Much as the current form this review is excellent, there are few identified concerns, which if addressed, will improve the overall quality of the article and make it even more attractive as well as suitable for readers
Weaknesses/Concerns:
1. 1, The sentence on line 181 should have a reference. Please, provide a citation for that sentence.
2 2. To be consistent, provide references for the pathogens indicated in the table 4
3. 3. To be consistent, provide references for information shown in the table 6
4 4. To be consistent, provide references for information shown in the table 7
5. Under RNA viruses as discussed from lines 372 to 418, construct a table similar to what you created for the DNA vaccines. That will show consistency.
Other concerns that should be addressed are shown below.
· None of the potential chitosan sides effects were discussed if chitosan-NPs and adjuvants are used in humans. If example, there are some clinical studies which suggest that if present in human as chitosan, it can interact with some clinical drugs including warfarin and acyclovir and lead to lead to other unwanted effects It also is known to cause nausea and stomach upset. It also may induce severe allergic responses in individuals who have allergy to shellfish and mushroom. Yet these were no discussed under the future uses
· As mentioned in the review that chitosan-NP delivery system or as well as its adjuvants could induce pro-inflammatory responses. Which proinflammatory cytokines are induced and does it also cause pro-inflammatory responses? Should discuss
· Is there evidence that chitosan-NP delivery system as adjuvant causes Th-17 responses? Discuss.
· It was not clearly discussed regarding whether the chitosan-based nanomaterial delivery system causes side effects in immunocompromised patients including diabetes patients. Discuss.
Round 2
Reviewer 1 Report (New Reviewer)
The alterations have improved the manuscript quality therefore I recommend the publication of this work.
Reviewer 2 Report (New Reviewer)
Thank you for responding positively to the reviewer and for implementing the reviewer's recommendations as well as including additional citations. These actions have significantly improved the overall quality of your review article on Chitosan nanomaterials.
This manuscript is a resubmission of an earlier submission. The following is a list of the peer review reports and author responses from that submission.
Round 1
Reviewer 1 Report
In this submitted review “Chitosan-based nanomaterial as immune adjuvant and delivery carrier in vaccine” the author has discussed about chitosan-based nanoparticles and their potential as adjuvant and delivery system.
This manuscript can be of interest for vaccine design and delivery; however, the manuscript is poorly written, and structured. It needs a major revision before publications.
1. The review structure needs to be drastically modified. The author should include introduction chitosan, chemical structure, modifications of chitosan done to be used as adjuvant and delivery needs to be addressed. How the modifications improve its efficiency needs to be addressed?
2. The abstract needs to be corrected. Restructure the sentences such as more and more studies. Line 16, line 20…
3. Change the sentence line 32 and include the references.
4. In mRNA vaccines, include the recent development of Covid-19 vaccines and how the delivery vehicle helped in successful delivery of mRNA vehicle in brief.
5. Restructure the sentence line 48, line 50.
6. Briefly, cite examples how different adjuvants have improved in vaccine efficacy.
7. Add more references in line 56.
8. Line 61, line 62, line 64, correct the sentences.
9. The author states their modification to chitosan, please explain the rational behind the modifications. List other modifications done in chitosan to enhance its activity.
10. Revise the sentences 71 and 72.
11. Line 80, the author states specific characteristics. Please clarify those specific attributes.
12. Please include recent development of viral vectors as vaccine delivery agent. Example J and J covid-19 vaccines.
13. Correct the sentences line 93…application limitation…
14. Restructure the sentences line 102. Chitosan-based delivery systems have been shown to possess mucoadhesive quality
15. Sentences Line 131 is constantly repeated throughout the manuscript.
16. Please explain how chitosan is acting as adjuvants and modifications like deacetylation, phosphorylation helped its adjuvant property.
17. Sentence 154 is too vague. Every nanoparticle function depends on particle size, zeta potential.
18. Preparations of chitosan-based particle, provide a schematic representation of these methods and the procedure for preparations. Also briefly explain which method is the best one?
19. Mechanism of chitosan as vaccine delivery system is explained poorly. Please explain concisely.
20. Figure 2, spelling errors for antigen. Change the legend bacterial viruses??
21. Sentence 184 is incomplete.
22. Line 201 Too much Th1.,line 256 needs corrections.
23. Line 340 correct it.
24. In the future perspective, include how optimization can be done for chitosan synthesis. How it can be a better adjuvant and delivery system compared to other? Provide details.
25. Correct Line 342. Meanwhiles, the application of DNA vaccine and mDNA vaccine are also increasing. Cite Covid-19 vaccine success..
Author Response
The point-by-point response to the reviewer’s comments can be seen in the attachment.

Reviewer 2 Report
Zhao et al. provided a systematic review of chitosan in vaccine research. This is a timely contribution, because research on adjuvant and vaccine delivery system now is proceeding at an unprecedented pace. There are many different adjuvants and delivery systems and many different applications. A focused and thorough review on a specific adjuvant or class of adjuvants will be helpful to the interested parties. The scientific content of this review is suitable for publication in Vaccines. However, the writing style and language is an issue. I strongly recommend the authors to get help from professional editing service. It needs a lot of editorial work.
Author Response

(The authors gave the same response as above.)

Reviewer 3 Report
This paper require significant rewriting before can be reviewed. I have impression that paper was written by undergrad student with very naïve understanding of the topic.
For example, just one paragraph:
“In many cases, vaccines only elicit too weak immunogenicity to prevent infection, thus, effective delivery system is required in vaccine preparation to enhance the immense effect [7]. Suitable vaccine adjuvant and vaccine delivery system can induce stronger immune responses to antigen and reduce the dosage and production cost in populations responding poorly to vaccination [2], hence, there is an urgent need to develop novel vaccine adjuvants and delivery systems to improve the immense effect of vaccine. With the development of material science and nanotechnology, it is becoming possible to rationally design and manufacture novel vaccine adjuvants and delivery systems with the required activity and safety.”
- vaccines only elicit too weak immunogenicity to prevent infection – such product is not vaccine (if could not prevent infection), the word “only” and “too” is also not really properly used in the sentence.
- immense effect – means what?
- reduce the dosage and production cost in populations responding poorly to vaccination – so in the other populations the dosage and cost are still high? What does exactly mean “reduce production cost in populations”, vaccines are produced in populations?
- improve the immense effect of vaccine – what is immense effect of vaccine? Define.
- Suitable vaccine adjuvant and vaccine delivery system can induce stronger immune responses ………. hence, there is an urgent need to develop novel vaccine adjuvants and delivery systems. – not nice style, repetition (minor)
- With the development of material science and nanotechnology, it is becoming possible to rationally design and manufacture novel vaccine adjuvants and delivery systems with the required activity and safety – How this is exactly related? Means once nanotechnology has been developed, we can produce whatever we want, for example cure for all cancers (with vaccines)? This is oversimplifying role of nanotechnology in the adjuvant design. Rational design of an adjuvant is rather related to discovery of the adjuvant mechanism of action, and not material science and nanotechnology.
Next paragraph, the same issues, plus practically each sentence is unrelated to previous one. General information is mixed with description of authors research data, vaccines are confused with delivery of antibodies, and “Chitosan-based nanoparticles have been used as adjuvant and delivery system for many viral and bacterial vaccines due to their small size” – vaccines have small size or nanoparticles? Moreover, all nanoparticles have small size (nanoparticles means nano-sized particles), high surface area etc., so why exactly chitosan was used?
Author Response

(The authors gave the same response as above.)

Round 2
Reviewer 1 Report
Commendable job. All comments are well addressed.